# Enhancing Polysulfone Mixed-Matrix Membranes with Amine-Functionalized Graphene Oxide for Air Dehumidification and Water Treatment

**DOI:** 10.3390/membranes13070678

**Published:** 2023-07-19

**Authors:** Omnya Abdalla, Abdul Rehman, Ahmed Nabeeh, Md A. Wahab, Ahmed Abdel-Wahab, Ahmed Abdala

**Affiliations:** 1Chemical Engineering Program, Texas A&M University at Qatar, Doha 23874, Qatar or o.abdalla@gord.qa (O.A.);; 2Gulf Organisation for Research & Development (GORD), Qatar Science & Technology Park, Tech1 Bldg, Suite 203, Doha 210162, Qatar

**Keywords:** functionalized graphene oxide, air dehumidification membranes, polysulfone membranes, mechanical properties, oil–water separation, membrane fouling

## Abstract

Porous low-pressure membranes have been used as active membranes in water treatment and as support for thin-film composite membranes used in water desalination and gas separation applications. In this article, microfiltration polysulfone (PSf)mixed-matrix membranes (MMM) containing amine-functionalized graphene oxide (GO-NH_2_) were fabricated via a phase inversion process and characterized using XPS, SEM, AFM, DMA, XRD, and contact angle measurements. The effect of GO-NH_2_ concentration on membrane morphology, hydrophilicity, mechanical properties, and oil–water separation performance was analyzed. Significant enhancements in membrane hydrophilicity, porosity, mechanical properties, permeability, and selectivity were achieved at very low GO-NH_2_ concentrations (0.05–0.2 wt.%). In particular, the water permeability of the membrane containing 0.2 wt.% GO-NH_2_ was 92% higher than the pure PSf membrane, and the oil rejection reached 95.6% compared to 91.7% for the pure PSf membrane. The membrane stiffness was also increased by 98% compared to the pure PSf membrane. Importantly, the antifouling characteristics of the PSf-GO-NH_2_ MMMs were significantly improved. When filtering 100 ppm bovine serum albumin (BSA) solution, the PSf-GO-NH_2_ MMMs demonstrated a slower flux decline and an impressive flux recovery after washing. Notably, the control membrane showed a flux recovery of only 69%, while the membrane with 0.2 wt.% GO-NH_2_ demonstrated an exceptional flux recovery of 88%. Furthermore, the membranes exhibited enhanced humidity removal performance, with a permeance increase from 13,710 to 16,408. These results indicate that the PSf-GO-NH_2_ MMM is an excellent candidate for reliable oil–water separation and humidity control applications, with notable improvements in antifouling performance.

## 1. Introduction

Water security is a global challenge caused by population growth, climate change, agriculture, and industrialization, resulting in water-stressed regions [1,2]. To address this challenge, innovative technologies for wastewater reuse are crucial [1]. Oily wastewater production has notably increased due to industrial activities like oil and gas extraction, food processing, and metal treatment [2]. In the US alone, daily production reaches volumes of 1.7 to 2.3 billion gallons [3]. Treating these oily water streams is vital to protect the environment and human health, as they are carcinogenic and harmful to ecosystems [4]. Various technologies, including membrane filtration, are being developed for oil–water separation [5]. Membrane technology offers high oil rejection rates and versatility in treating a wide range of compositions [6].

Porous membranes, such as polysulfone (PSf), are commonly used in oil–water separation and as a support for air dehumidification membranes [7,8]. However, the water vapor permeance (WVP) performance of existing porous supports, like PES and PSf, is limited [9,10,11,12]. Ultrafiltration polymeric membranes, particularly PSf, have demonstrated good performance in oil–water separation [13,14,15]. Additionally, membranes have been proposed as exciting candidates for humidity control by separating water from air [7,8]. To further optimize their reliability and reduce cleaning costs, incorporating nanomaterials into the polymer matrix is an effective approach [10]. These composite membranes, known as mixed-matrix membranes (MMMs), enhance permeability, mechanical stability, rejection, and antifouling properties.

Two-dimensional (2D) graphene oxide (GO) and its derivatives are promising nanomaterials for membrane modification. GO has an sp^2^ carbon honeycomb structure similar to graphene but contains oxygen functional groups such as hydroxyl (–OH), epoxy (C–O–C), carbonyl (C––O), and carboxylic (–COOH) [16]. GO has remarkable properties like high reactivity and solubility, ease of synthesis, and a very hydrophilic nature [17,18]. These properties make GO one of the most researched dope nanomaterials for membrane modification [19,20,21,22,23]. However, GO alone may not be sufficient to achieve the desired performance of the membranes, as it may suffer from leaching, aggregation, or instability. Therefore, various strategies have been developed to functionalize GO with different molecules or nanoparticles to enhance its properties and interactions with the polymer matrix [24,25].

Hwang et al. reported the fabrication of PSf membranes coated with a hydrophilic layer of GO. A phase inversion process first fabricated the PSf membranes. The prepared PSf/GO membrane improved hydrophilicity, permeability, and antifouling performance. The contact angle of the PSf/GO membrane decreased by 20.5° compared to the pure PSf membrane. Moreover, the flux recovery after fouling of the PSf/GO improved from 85.6% for PSf membrane to 90.4% for PSf/GO membrane [26]. However, these PSf/GO membranes suffer from leaching and delamination of the coated GO layer from the membrane surface, limiting their potential application [27]. PEBA MMMs with GO also face some additional challenges, such as aggregation and uneven dispersion of GO sheets, and weak interfacial adhesion between GO and the PEBA matrix [28]. Functionalized nanofillers or a coating material help to improve the hydrophilicity, antifouling, and antibacterial properties of membranes [25,29], which is important for their longevity.

Luque-Alled and coworkers [30] incorporated various concentrations of GO functionalized with 3-aminopropyltriethoxysilane (APTS) (APTS-GO) into the matrix of polyethersulfone (PES) and fabricated a MMM via non-solvent induced phase inversion (NIPS). The membrane porosity was significantly decreased upon adding APTS-GO, changing the membrane type from UF to NF. Additionally, a higher water flux of 9.9 L/m^2^hr Bar (LMH/bar) was achieved using the 0.1 wt.% APTS-GO membrane compared to 0.5 LMH/bar for the pure PES membrane. The performance of the membrane for rejection of magnesium sulfate (MgSO_4_), sunset yellow dye (SY), acridine orange dye (AO), and bovine serum albumin (BSA) was analyzed. The achieved rejection was 96.5%, 97.4%, 96.5%, and 51.6% for BSA, SY, AO, and MgSO_4_, respectively. Furthermore, improved antifouling and mechanical characteristics were obtained upon incorporating APTS-GO.

Zambare et al. [31] studied the impact of the amine chain length on coupling-agent-assisted GO functionalization with ethylenediamine, diethylenetriamine, and triethylenetetramine. The functionalized GO was then incorporated into the PSf solution to fabricate MMMs and investigate their performance. The functionalized GO membranes’ characteristics and performances were improved in terms of pure water flux, which reached 170.5 LMH/bar compared to 56.1 LMH/bar for the control membrane. The highest performances were attained by the 1.0 wt.% of EDA/GO membrane, as EDA reacted the most during the reaction, achieving the highest conversion among other amines. Additionally, the 1 wt.% EDA/GO membrane showed the best antifouling characteristics, as illustrated by it having the highest BSA flux recovery. However, the method used to functionalize GO with EDA is complicated and would face some difficulties during scale-up.

In this study, we present a novel and facile approach to functionalize graphene oxide with ethylenediamine (EDA), a simple and scalable coupling agent, and incorporate it into the PSf matrix to fabricate mixed-matrix membranes (MMMs) with superior performance for oil–water separation and humidity control applications. We use a versatile and cost-effective phase inversion process to produce asymmetric porous microfiltration membranes with enhanced interfacial adhesion and reduced aggregation of GO sheets. We investigate the effect of EDA-functionalized graphene oxide (GO-NH_2_) concentration on membrane morphology, hydrophilicity, mechanical properties, and oil–water separation performance. We also evaluate the antifouling characteristics of the PSf-GO-NH_2_ MMMs by filtering the bovine serum albumin (BSA) solution. Furthermore, we assess the potential of the PSf-GO-NH_2_ MMMs as multifunctional and reliable porous supports for air dehumidification membranes by measuring their water vapor permeance and water/air selectivity. Our results indicate that the PSf-GO-NH_2_ MMMs are excellent candidates for oil–water separation and humidity control applications.

This study uses a facile functionalization of GO with ethylenediamine (EDA) to synthesize amine functionalized GO (GO-NH_2_). Different GO-NH_2_ was incorporated into the dope solution of PSf, and PSf-GO-NH_2_ MMMs were fabricated via phase inversion. The influences of GO functionalization on the characteristics (porosity, hydrophilicity, roughness, and mechanical properties) and performance (water permeability, oil rejection, and antifouling properties) of the prepared MMMs were investigated. A fouling resistance test was performed by filtering a model BSA protein through the membrane. The structural and chemical changes induced by the incorporation of GO-NH_2_ into the PSf matrix were analyzed and correlated with membrane performance. Additionally, the performance of the membranes was assessed as a porous support for air dehumidification applications. Finally, we measured the water vapor permeability (WVP) and the H_2_O/N_2_ selectivity of the original and modified membrane to evaluate how well they support the thin-film composite (TFC) air dehumidification membranes.

## 2. Materials and Methods

### 2.1. Materials

Graphene oxide (GO; SE2430) from the sixth element was used as received. Ethylenediamine (EDA), Methanol (MeOH), Dimethylacetamide (DMAc), Polysulfone (PSf) (Mn ~ 22,000), Polyvinylpyrrolidone (PVP) (Mw ~ 55,000), Acetone, and Bovine Serum Albumin (BSA) (Sigma Aldrich, St. Louis, MO, USA) were received and used without further purification. Sodium hydroxide (NaOH) was obtained from Research-Lab, diesel was obtained from Woqod petrol station, and Deionized water (DI) was also used in this study.

### 2.2. Functionalization and Characterization of GO with EDA (GO-NH_2_)

A total of 900 mg of GO was dispersed in 300 mL of DI water via bath sonication for 4 h. Then, 1.6 mL EDA was added to the GO solution, and the functionalization reaction was carried out for 4 h at 85 °C under reflux. The reaction mixture was then cooled to room temperature, filtered, and repeatedly washed with DI water and MeOH until reaching pH 7. The morphology of GO and GO-NH_2_ was analyzed using SEM (FEI Quanta 400 SEM, Thermo Fisher Scientific, Waltham, MA, USA) and TEM (FEI TECNAI G2 TF20, Thermo Fisher Scientific, USA). An X-ray diffractometer (XRD: Ultima IV, RIGAKU, The Woodlands, TX, USA) with Cu-α radiation (λ = 0.15418 nm) operating at 40 kV and 20 mA was used to estimate the stacking characteristics and interlayer d-spacing of GO and GO-NH_2_ using Bragg’s law [32]. The composition and the functional groups on the surface of GO and GO-NH_2_ were investigated using Escalab 250 Xi (Thermo Fisher Scientific, USA) X-ray photoelectron spectroscopy (XPS) with 20 eV pass energy for high-resolution scans and 100 eV for the survey scans.

### 2.3. Membrane Fabrication

Phase inversion was employed to fabricate the UF PSf and PSf-GO-NH_2_ MMMs [33]. A mixture of PSf (15%) and PVP (5%) as a pore-forming agent were dissolved in DMAc (80%) by stirring at room temperature to prepare the control polymer solution. The concentration of the filler GO-NH_2_ varied from 0% to 0.8% (0%, 0.05%, 0.1%, 0.2%, 0.4%, and 0.8%) to prepare multiple membranes. The preparation process was carried out as follows: the membrane with the highest filler concentration, i.e., 0.8 wt.% GO-NH_2_ (relative to PSf), was made by dispersing GO-NH_2_ (60 mg) into 12.5 g of DMAc via probe sonication for 5 min. The f-GO dispersion in DMAc was then added to the dope polymer solution (7.5 g of PSf, 2.5 g of PVP, and 37.5 g of DMAc), and the mixture was subjected to bath sonication for 1 h, followed by probe sonication for 5 min and degassing for 20 min. The remaining membrane solutions with lower GO-NH_2_ concentrations were prepared by mixing the 0.8% solution with the dope solution polymer solution. The six polymer solutions were then cast on a glass plate using an automated membrane-casting machine (Promotor MemCast) running at a speed of 4 m/min and a casting knife 150 μm in thickness. After 30 s, the glass plate was then immersed in a coagulation bath containing DI water for 5 min. The cast membrane sheets were rinsed with DI water with daily water changes for five days before testing to remove PVP.

### 2.4. Membrane Characterization

The membrane surface topography and cross-section morphology were analyzed using SEM (FEI Quanta 400 SEM, Thermo Fisher Scientific, USA). For membrane surface analysis, square samples were cut and mounted on the sample holder using carbon tape, while for the cross-sectional images, the membrane samples were cryo-fractured in liquid nitrogen. The hydrophilicity of the membranes was estimated by measuring the water contact angle (WCA) of the membrane via the sessile drop method using a Kruse drop shape analyzer (KDSA: DSA25, Germany) [34]. The membranes’ surface roughness was analyzed using Atomic Force Microscopy (AFM) (Dimension Icon, Bruker, Billerica, MA, USA) operating in tapping mode using 50 × 50 µm scan size. The mechanical properties of the membranes, such as Young’s modulus, breakage strength, and breaking strain, were measured via uniaxial tensile testing using dynamic mechanical analysis (DMA: Q800, TA Instruments, New Castle, DE, USA). The overall membrane porosity was measured using the wet/dry weight method. The (%) porosity was defined as a function of weight using the following equation:(1)$$\mathrm{Porosity}(\%)=\frac{W_{w}-W_{d}}{\rho_{w}V}\times 1$$
where $W_{d}$ and $W_{w}$ are the weight of the dry and wet membrane (g), respectively, $\rho_{w}$ is the density of pure water (g/m^3^), and V is the membrane volume.

### 2.5. Membrane Performance and Antifouling Characteristics

A dead-end cell (Sterlitech HP4750, Kent, WA, USA) was used to estimate the membrane’s permeability by measuring the water flux through the cell at trans-membrane pressures 1, 2, and 3 bars. The flux was calculated using the following equation:(2)$$J=\frac{m}{t*A}\times 0.06\;(L/m^{2}\mathrm{hr})$$
where J is the pure water flux (L/m^2^hr), (LMH)), m is the mass of water permeated (g), t is the time (minutes), and A is the active membrane area (14.6 cm^2^).

The membrane selectivity was evaluated by measuring oil (diesel) rejection. First, 1000-ppm oil emulsion was prepared as a stock solution via probe sonication. The stock solution was further diluted to 100-ppm oil emulsion and used as the feed. Three permeated samples and one residual sample were collected. The carbon content of the feed, the permeate samples, and the residual sample was analyzed using a total organic carbon analyzer (TOC: TOC-L, Shimadzu, Kyoto, Japan). The percentage of oil rejection was calculated according to the following equation:(3)$$R\left(\%\right)=\frac{\left(C_{F}-C_{p}\right)}{C_{F}}*100$$
where C_p_ and C_F_ are the TOC concentrations of oil permeate and feed.

To study the membrane resistance to organic fouling, 200 mL of a 500 ppm BSA solution was permeated through the control membrane and one selected MMM at 1 bar. After the permeation experiment was completed, the membrane was cleaned by being rinsed with DI water twice, flushed with 2 M NaOH solution, and reused for an additional cycle.

To study the tradeoff between permeability and selectivity, separation and permeability factors were calculated as follows:(4)$$SF=\frac{1}{1-R}$$
(5)$$PF=\frac{J}{1000}$$
where R is the fraction of oil rejection, and J is the pure water flux.

### 2.6. Water Vapor Permeance and Water/Air Selectivity

The air dehumidification performance of the membrane was tested at specific relative humidity levels and selected inlet feed pressures. The setup comprised an air dehumidification module with a humidity controller, following the process developed by Culp [35]. Humidified inlet air entered the membrane cell, which released dry air and allowed water vapor to pass through the permeate side (facilitated by a vacuum pump), as shown in Figure 1.

High-purity air (>99.9%), obtained from the National Industrial Gas Plant, was utilized as a carrier for the water. The absolute humidity was meticulously set to approximately 0.027–0.028 kg/m^3^, corresponding to 85–90% humidity. The air flow rate was regulated at precisely 2 SLPM, as measured by an Alicat Scientific mass flow meter. The temperature and relative humidity were ascertained using highly accurate sensors from Vaisala Oyj, boasting an accuracy of 1.0%. Both the retentate and permeate flow rates were determined using a mass flow meter, while a pressure transducer from Gentek facilitated the measurement of pressure at both locations. Lastly, the permeate pressure was maintained at a consistent 2 mBar (achieved with an IKA Vacstar pump). The permeance equation is a mathematical relationship that relates the gas flow rate through a membrane to the pressure difference across the membrane. It is commonly used to calculate the permeance of a membrane to a particular gas or vapor.
(6)$$P_{i}=\frac{N_{i}}{A\Delta p_{i,lm}}$$
(7)$$\Delta p_{i,lm}=\frac{\Delta p_{i,1}-\Delta p_{i,2}}{\ln\left(\Delta p_{i,1}/\Delta p_{i,2}\right)}$$

P_i_ is the permeance in (mol/m^2^ s Pa), N_i_ is flow rate of component i (H_2_O or air) across the membrane (mol/s), A is the membrane area (m^2^), Δp_i,lm_ is the log mean partial pressure difference of component i across the membrane (Pa), Δp_i,1_ is the partial water pressure difference across the membrane at the feed side (p_i,f_ − p_i,P_), and Δp_2_ is the pressure difference across the membrane at the retentate side (p_i,R_ − p_i,P_).

## 3. Results and Discussion

### 3.1. Characterization of GO and GO-NH_2_

Figure 2a,b shows SEM images of the GO and GO-NH_2_ samples. The morphology of GO and GO-NH_2_ appears to be almost the same with multiple layers, revealing folded structures with distinct wavy features and edges, which is consistent with the literature [36]. Comparing GO and GO-NH_2_, the functionalized GO appears to have a more wrinkled and folded structure than GO. However, the structure is retained even after functionalization. A single GO sheet can be seen in Figure 2d when focusing on the TEM image’s light regions, whereas dark regions indicate multi-folds and layers.

XRD was used to estimate the stacking characteristics and crystal structure of GO and GO-NH_2_ samples, and the XRD patterns in the 2θ range of 5° to 35°, are shown in Figure 2c. A clear dominant peak at 2θ = 9.42° was present in the GO pattern. This peak is consistent with the previously reported GO (002) peak [37,38]. For the GO-NH_2_ sample, the (002) peak’s intensity was significantly reduced, broadened, and shifted to the right, suggesting the exfoliation of graphite oxide structure. The d-spacing values of the GO and GO-NH_2_ samples, calculated from Bragg’s Law, were 9.38 Å and 8.50 Å, respectively. The reduction in the d-spacing of the GO-NH_2_ peak is attributed to the functionalization of GO with EDA, as the amine functional groups brought the interlayer space between GO sheets together, enhancing the stability of GO.

The elemental compositions of GO and GO-NH_2_ were investigated using EDX. Supplementary Information Appendix A shows the atomic composition of GO and GO-NH_2_. It is shown that GO is composed only of carbon and oxygen, with 63.15% and 36.85%, respectively [39]. At the same time, GO-NH_2_ contains 7.35% N in addition to carbon and oxygen. The presence of nitrogen in the GO-NH_2_ sample confirms the successful functionalization of GO with EDA since EDA has amine (NH_2_) functional groups in its structure. The elemental mapping of different elements, i.e., carbon, oxygen, and nitrogen on the surface of GO-NH_2,_ is shown in Supplementary Information Appendix A. The homogenous distribution of the organic functionality on the surface of GO-NH_2_ is observed, where carbon, oxygen, and nitrogen are spread evenly.

Furthermore, XPS analysis was employed to investigate the bonding and chemical composition of GO before and after functionalization. Figure 3a shows the XPS elemental surveys for GO and GO-NH_2_ across binding energies from 100 eV to 900 eV. GO is reiteratively confirmed to consist only of carbon and oxygen with traces of nitrogen, as shown by the two dominant peaks at 283 eV and 533 eV for carbon and oxygen, respectively (similar to the peaks reported by Sali et al.) [40]. A new peak at 399 eV, corresponding to nitrogen, appears in the GO-NH_2_ survey with a nitrogen content of 7.2% for the GO-NH_2_, confirming the successful functionalization of GO with EDA. The high-resolution scans of C_1s_ and N_1s_ are shown in Figure 3b–d. The high-resolution deconvolution of the C_1s_ scans, considering sp^2^ carbon (C_1s_) graphene, 283.63 eV) (C–C, 284.74 eV) groups, epoxy (C–O, 286.75 eV) groups, and carbonyl (C=O, 288.3 eV) groups are observed. A critical observation in the deconvoluted bonds was the drastic decrease in the epoxy (C–O) atomic composition (C–O) bond from 55.1% in GO to 26.8% in GO-NH_2_. This outcome occurred due to the amine functional groups replacing the epoxy groups during the functionalization of GO with EDA, along with the clear presence of C-NH_2_ groups in the N_1s_ deconvoluted spectrum.

### 3.2. Membrane Characterization

The surface and cross-sectional morphology of the pure PSf membrane and the MMMs containing 0.2 and 0.8 wt.% GO-NH_2_ were analyzed using SEM, and the results are shown in Figure 4. The surface of the control membrane is smoother and less porous than MMMs membrane. Moreover, the cross-sectional morphologies of the mebranes exhibit the typical finger-like channeled sublayer expected for asymmetric porous membranes. However, these highly porous finger-like channels are covered by a well-structured dense skin layer in the MMMs (Figure 4e,f). Similar morphology was reported by Zinadini et al. in a previous study for a system of polyethersulfone (PES) mixed-matrix nanofiltration membrane containing graphene oxide (GO) nanoplates [41]. The sub-layer morphology can be described as finger-like micro-voids with sponge features at the bottom; this layer gives the needed mechanical stability for the fabricated membranes. The skin layer is the side that is open to air when casting, while the sub-layer is the side that is in contact with the glass plate. Comparing the control membrane with GO-NH_2_ membranes, the separating layer of the control membrane appears to be almost non-existent, while a much thicker separating layer is found in the GO-NH_2_ membranes, indicating higher separation possibilities.

Moreover, the sub-layer of the 0.2 wt.% GOO-NH_2_ MMM is thicker, more stretched, and possesses the thickness of finger-like voids more so than the control membrane. This could be due to the very hydrophilic NH_2_ groups on the surface of the membrane, which enhances and quickens the transfer between the DMAc and DI water during phase inversion, thus increasing the pores and channel size [42].

The surface hydrophilicity of the prepared membranes was tested by measuring water contact angles (WCAs). The test was performed on a square sheet sample of membranes to measure the wettability variation due to the various GO-NH_2_ concentrations studied. High water contact angles imply that the membranes have greater hydrophilicity and vice versa. The membrane with the highest hydrophobicity was the control membrane, with 77° WCA, consistent with previously reported measurements [31]. On the other hand, the highest hydrophilicity was attained by the 0.2 wt.% GO-NH_2_ membrane, with a reduction of almost 14° compared to the control membrane. Figure 5a shows the variation in WCA as the concentration of GO-NH_2_ increases. At low concentrations, the contact angle decreases to a minimum, increasing again at higher concentrations. This trend was observed because low concentrations of GO-NH_2_ have a strong attraction to water during phase inversion. Consequently, GO-NH_2_ moves toward the membrane/water interface, facilitating a low-energy interface that results in high hydrophilicity [43].

The fabricated membranes’ three-dimensional (3D) morphologies and respective surface roughness values were determined using AFM under tapping mode. A scan size of 50 × 50 µm was chosen for this study. Figure 5b shows the variation in surface roughness among the different concentrations of GO-NH_2_, along with selected topographical AFM images. The three selected AFM images illustrate the distinct extent of roughness that expresses the variation among all GO-NH_2_ concentrations. Dark regions in the AFM images define deep points such as pores and valleys, while light-colored regions define higher points such as peaks; the exact elevation scale can be seen on the right side of the image [41]. The main physical parameter obtained from AFM analysis is the average surface roughness (Ra), which describes the standard surface deviation of the designated surface, hence calculating the roughness. The roughness variation followed a trend whereby, at small loadings, the roughness increases to a maximum and decreases back at high loadings. This trend is consistent with that previously reported by Mokkapati and coworkers [44]. The MMMs are believed to be more porous than the control membrane, increasing the roughness upon adding GO-NH_2_, implying that the smoothest membrane is the control membrane. The membrane with the highest roughness was found to be 0.1 wt.% GO-NH_2_, with a roughness of 62.05 nm compared to 51.35 nm for the control membrane. Other AFM 3D images are provided in the Appendix A.

The application of the prepared MMMs greatly depends on their mechanical characteristics. DMA was applied to the fabricated membranes to investigate the effect of GO-NH_2_ on mechanical properties. Three specimens were tested from each membrane sample, as shown in Figure 6a. Moreover, Young’s modulus and breakage strength were calculated to fully understand the mechanical properties of the membranes. Figure 6b illustrates the effect of GO-NH_2_ as a filler on Young’s modulus and breakage strength. The 0.2 wt.%. membrane attained the highest Young’s modulus. GO-NH_2_ membrane was about 116.67 MPa. When GO-NH_2_ was added, the stiffness and strength of the material increased to reach a maximum at 0.2% by weight. Beyond that point, the material became weaker, but it was still stronger than the material without GO-NH_2_. This trend is explained by the fact that low-concentration GO-NH_2_ membranes possess a high aspect ratio and excellent interfacial adhesion between the GO-NH_2_ and the polymer, facilitating the development of their outstanding mechanical properties [45]. However, at high concentrations, the GO-NH_2_ filler aggregate in the polymer matrix reduces the mechanical properties [46,47]. The optimum mechanical properties were attained by the 0.2 wt.%. membrane. The GO-NH_2_ membrane’s Young’s modulus and breakage strength were improved by 98% and 50% compared to the control membrane. The steepest slope and high stress of the 0.2 wt.%.

The wet/dry method was used to calculate the porosity of the prepared membranes. Figure 7 shows the variation in the membrane’s porosity as the concentration of the GO-NH_2_ filler changes. The incorporation of GO-NH_2_ led to an increase in porosity to a maximum of 0.2 wt.% GO-NH_2_ followed by a decrease at higher concentrations. The increase in porosity was directly related to the hydrophilicity of the PSf-GO-NH_2_ mixture, which accelerated the exchange between the solvent and non-solvent. The highest enchantment in the porosity was about 11.39%, from 57.52% for the control membrane to 68.91% for the 0.2 wt.% GO-NH_2_ membrane.

### 3.3. Performance Testing of the GO-NH_2_ Membranes

The performance of the membranes was tested by checking the permeability, selectivity, and antifouling characteristics. The permeability test involved checking DI water flux at a trans-pressure of 1, 2, and 3 bars using a dead-end cell. Before testing, the prepared membranes were pressurized at various pressures from 1 to 7 bars to ensure stable operation. Figure 7 shows the average pure water fluxes at different concentrations of GO-NH_2_. It can be observed from the figure that pure water fluxes increased when small amounts of GO-NH_2_ (0.05–0.2 wt.%) were added and decreased again at more significant amounts (0.4–0.8 wt.%). The same trend was observed by Zhao et al. and Abdalla et al. in one of their previous studies [47,48]. This result is associated with the agglomeration of the GO-NH_2_ filler when present in large amounts, as it negatively affects the porosity and hydrophilicity. The high concentration of the GO-NH_2_ filler increases the dope solution viscosity, which hinders the exchange between solvents and non-solvents during phase inversion, inducing a decrease in hydrophilicity and porosity. Therefore, the pure water flux decreases [42]. The 0.2 wt.% GO-NH_2_ membrane achieved the highest permeability (92% higher than the control membrane).

The extent of oil–water separation was used to assess the selectivity of the fabricated membranes. A 100-ppm oil–water mixture was filtered through the prepared membranes, and the percentage of oil rejection was calculated. Figure 8a shows the percentage of oil rejection for the control and GO-NH_2_ membranes. As demonstrated by Figure 8a, the oil rejection improved for all MMMs, except for the membrane with the highest GO-NH_2_ concentration. The control membrane had an oil rejection of 91.7%, while the 0.2 wt.% GO-NH_2_ membrane exhibited an oil rejection of 95.6%, with an improvement of about 4%; consequently, the oil content of the 0.2 wt.% GO-NH_2_ permeate (4.4 ppm) was much lower than that of the control membrane (8.3 ppm), indicating greater separation efficiency [49].

Generally, there is a tradeoff between the permeability and selectivity of polymeric membranes, as higher water fluxes usually tend to reduce the separation efficiency [50]. This study examined this tradeoff by plotting the separation factor vs. permeability factor for all prepared membranes, as presented in Figure 8b. The separation and permeability factors were calculated using the equations presented in Section 2.4. This study found the tradeoff to be minimal, as the highly permeable membranes exhibited high separation factors. Therefore, incorporating small quantities of GO-NH_2_ positively influenced the fabricated membranes’ performance in multiple criteria. The optimum membranes were found to be 0.05 wt.%, 0.1 wt.%, and 0.2 wt.% GO-NH_2_, as highlighted by the light green color in Figure 8b, and the poorest performing membranes were the control and 0.8 wt.% GO-NH_2_ membranes.

Fouling is one of the main issues limiting the practical application of membranes since it controls the membrane performance over a period of time [51]. Membrane fouling is caused by either pore blocking, cake formation, organic adsorption, inorganic precipitation, or biological fouling, resulting in short-term or immutable flux decline [52]. In this study, an organic fouling test was performed by filtering 500-ppm BSA solution through the membranes, followed by rinsing with DI water, 0.1 M NaOH solution, and a second BSA cycle.

Each step measured the BSA flux to check the fouling behavior in flux decline and recovery ratio. An antifouling test was only carried out on the control and 0.2 wt.% GO-NH_2_ membrane since it was the optimum membrane among the rest in terms of permeability, selectivity, and mechanical properties. The two membranes were pre-compacted at high pressures before BSA filtration to ensure that the flux decline was only caused due to the fouling phenomenon. Figure 9 shows the variation in BSA fluxes with volume for the control and 0.2 wt.% GO-NH_2_ membrane. As demonstrated by the figure, the control membrane exhibited the highest flux reduction compared to the modified membrane. This is associated with the low hydrophilicity of the control membrane compared to the 0.2 wt.% GO-NH_2_ membrane, indicating higher fouling potential. Therefore, from the results, it can be deduced that the incorporation of GO-NH_2_ resulted in improved antifouling characteristics due to the higher achieved hydrophilicity. Furthermore, as investigated through the second BSA fouling step, the modified membrane displayed enhanced flux recovery compared to the control membrane. The control membrane showed a flux recovery of only 69%, while a flux recovery of 88% was recorded for the 0.2 wt.% GO-NH_2_ membrane (Table 1).

### 3.4. Performance Testing of the GO-NH_2_ Membranes for Air Dehumidification

The incorporation of 0.20 wt.% GO-NH_2_ into the PSf matrix enhances the water vapor and air permeance of the PSf-GO-NH_2_ membranes, as shown in Table 2. This is attributed to the following factors: (1) GO-NH_2_ increases the porosity of the membranes, which reduces the transport resistance and increases the mass transfer rate of water vapor and air molecules through the membranes; (2) GO-NH_2_ increases the membrane hydrophilicity (decreases the water contact angle), which increases the affinity and adsorption of water vapor molecules on the membrane surface and within the pores. These factors are beneficial for air dehumidification applications, where high water vapor permeance is desirable. However, the increased porosity also has a negative effect on the selectivity and separation factor of the membranes. This is because both larger pores and a higher porosity allow for more non-selective gases, such as nitrogen and oxygen, to pass through the membranes, along with water vapor, reducing the relative difference in permeance between water vapor and other gases. Selectivity and separation factor are important parameters for evaluating membrane performance for gas separation applications, where high gas purity is required. The water vapor permeance of the membranes with 0.20 wt.% GO-NH_2_ was 16,408 GPUs, which was ~20% higher than that of the pure PSf membrane, while the selectivity and separation factor were slightly lower. The water vapor permeance of the PSf-GO-NH_2_ membranes is higher than that of other PSf-based membranes [53,54], which makes them ideal candidates as substrates for thin-film composite (TFC) air dehumidification membranes. This study highlights the first use of amine-functionalized graphene oxide (GO-NH_2_) as an additive to enhance the features of PSf matrix, as GO-NH_2_ offers advantages other graphene-based additives, such as improved dispersion, compatibility, and functionalization [55].

## 4. Conclusions

In conclusion, a facile functionalization method was developed to functionalize GO with amine groups. The functionalization was confirmed through multiple characterization techniques such as XRD, SEM, EDX, and XPS. Amine-functionalized GO (GO-NH_2_) was incorporated into the polymeric mix via straightforward addition to the casting solution. The prepared membranes, i.e., PSf/GO-NH_2_, were fabricated via phase inversion. The influence of GO-NH_2_ on the morphology, hydrophilicity, mechanical properties, performance, and antifouling characteristics of the membranes was further investigated through SEM, contact angle measurements, pure water flux, and oil rejection. The addition of GO-NH_2_ into polysulfone matrix enhanced the hydrophilicity and mechanical properties of the membranes since GO-NH_2_ has a hydrophilic nature and a high aspect ratio. Outstanding improvements in mechanical properties were demonstrated through the high Young’s modulus and breakage strength values attained by the MMMs.

Furthermore, the prepared membranes’ permeability and selectivity were elevated by almost 91.7% and 5.08% compared to the control membrane. Moreover, a robust antifouling capability was observed for the 0.2 wt.% GO-NH_2_ membrane because of its high hydrophilicity and electrostatic repulsion characteristics. Consequently, the 0.2 wt.% GO-NH_2_ membrane was chosen as the optimum membrane as it showed the best performance and properties among all prepared membranes, as shown in Table 1. Finally, further investigations must be carried out to better understand the specific interactions between water and GO-NH_2_ membranes. The initial results show tremendous potential and indicate that these MMMs could be utilized in various industrial applications. Regarding the membrane performance for air dehumidification, the addition of GO-NH_2_ to the membrane increased the water vapor permeance by 20%, indicating an enhancement of the membrane air-dehumidification characteristics.

## Figures and Tables

**Figure 1 membranes-13-00678-f001:**
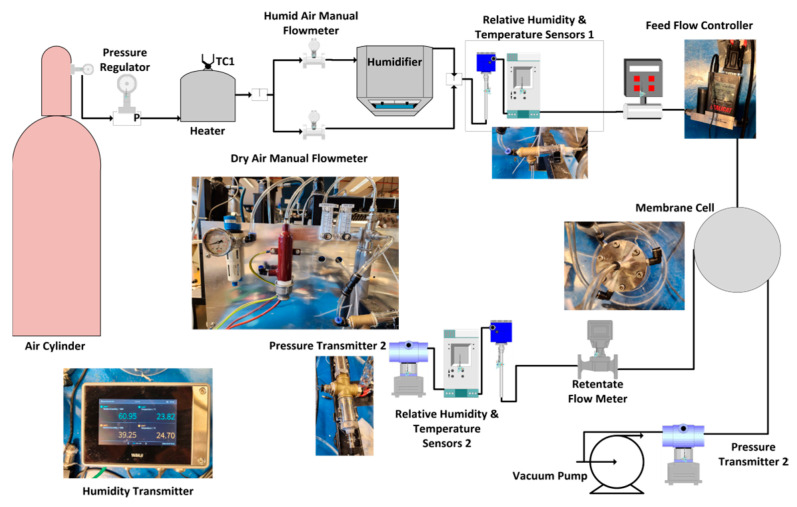
Testing setup for our vacuum-based air dehumidification performance analysis.

**Figure 2 membranes-13-00678-f002:**
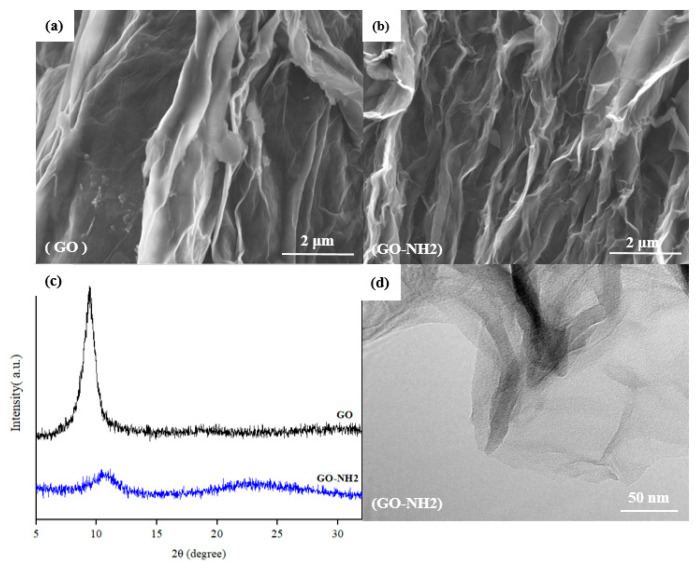
Characterization of GO and GO-NH_2_: (**a**) SEM Images of GO, (**b**) SEM Images of GO-NH_2_, (**c**) XRD Patterns of GO and GO-NH_2_, and (**d**) TEM Image of GO-NH_2_.

**Figure 3 membranes-13-00678-f003:**
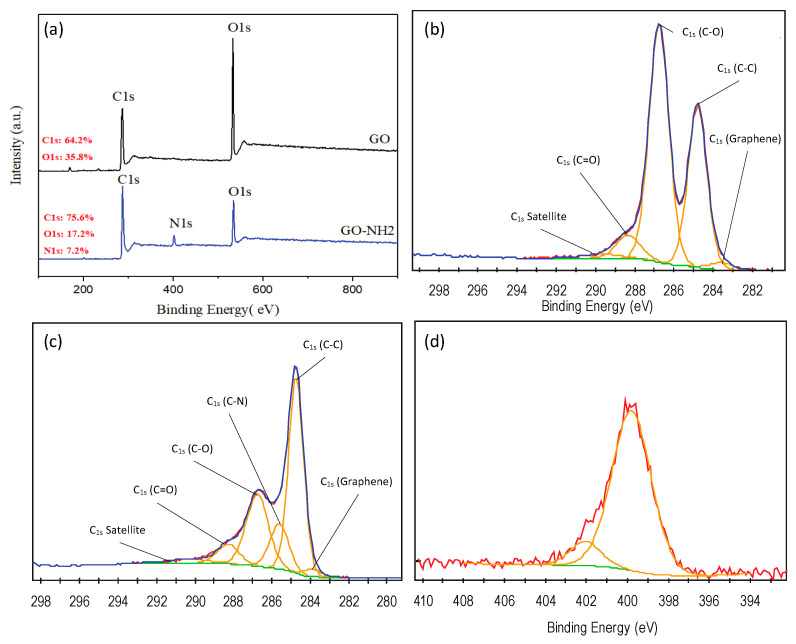
XPS Analysis of GO and GO-NH_2_: (**a**) XPS Surveys of GO and GO-NH_2_, (**b**) XPS C_1s_ Deconvoluted Spectra of GO, (**c**) XPS C_1s_ Deconvoluted Spectra of GO-NH_2_, and (**d**) XPS N_1s_ Deconvoluted Spectra of GO-NH_2_.

**Figure 4 membranes-13-00678-f004:**
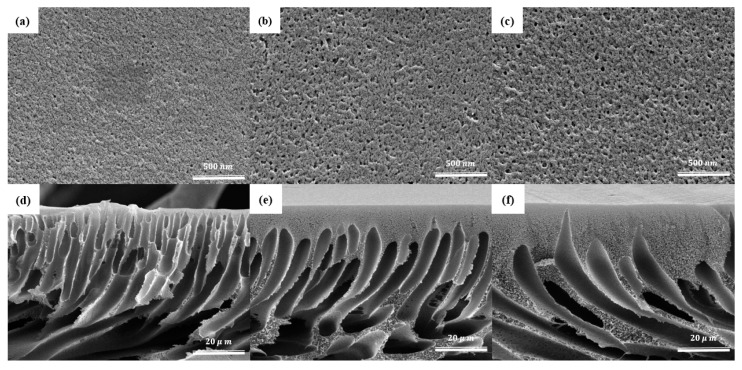
Surface SEM images of GO-NH_2_ membranes—(**a**) control, (**b**) 0.2%, (**c**) 0.8%—and cross-section SEM images of GO-NH_2_ membranes with (**d**) 0%, (**e**) 0.2%, and (**f**) 0.8% GO-NH_2_.

**Figure 5 membranes-13-00678-f005:**
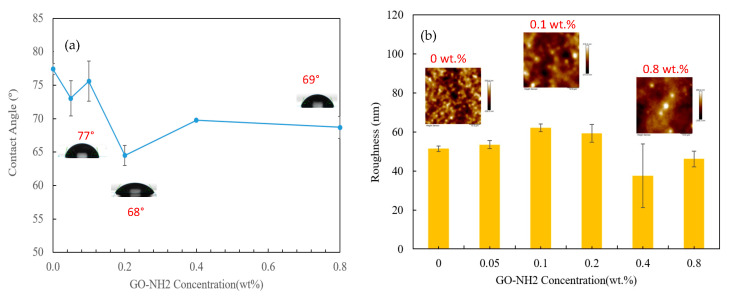
Effect of GO-NH_2_ concentration on (**a**) Water Contact Angle and (**b**) Roughness of PSf-GO-NH_2_ MMM.

**Figure 6 membranes-13-00678-f006:**
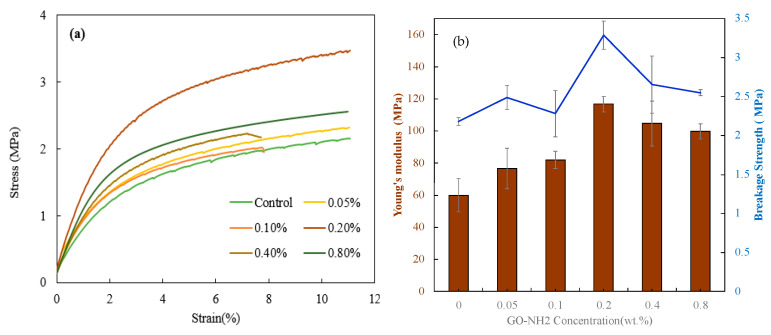
Effect of GO-NH_2_ concentration on (**a**) Young’s modulus and fracture strength of the PSf/GO-NH_2_ MMMs and (**b**) Young Modulus (Brown Bars) and Breakage Strength (Blue Curve). Stress vs. strain for the GO-NH_2_ membranes.

**Figure 7 membranes-13-00678-f007:**
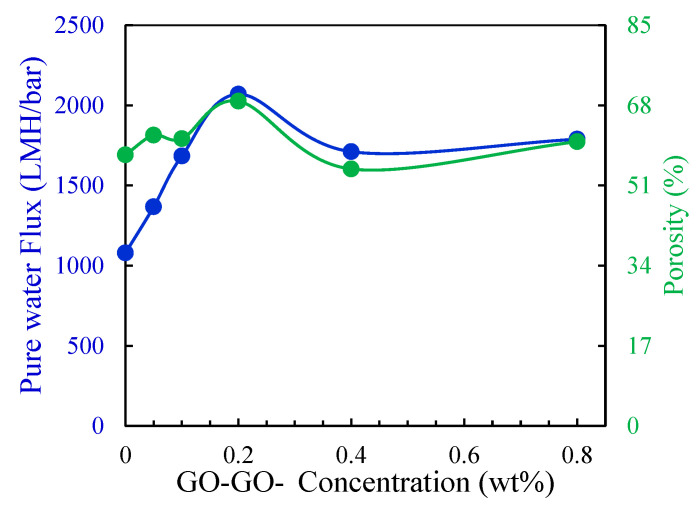
Porosity and water fluxes of control and GO-NH_2_ membranes.

**Figure 8 membranes-13-00678-f008:**
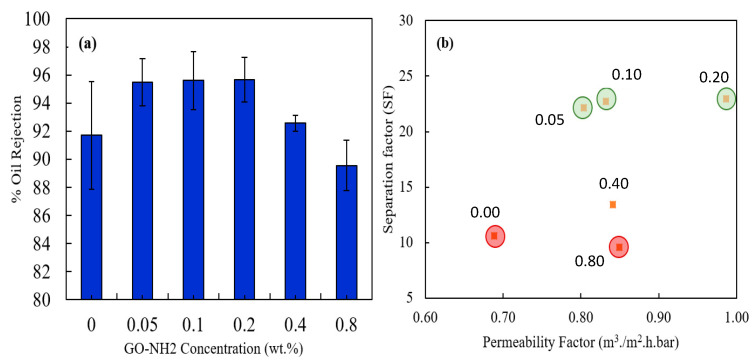
Comparison of GO-NH_2_ Membranes: (**a**) Percentage of Oil Rejection and (**b**) Correlation between Separation Factor and Permeability.

**Figure 9 membranes-13-00678-f009:**
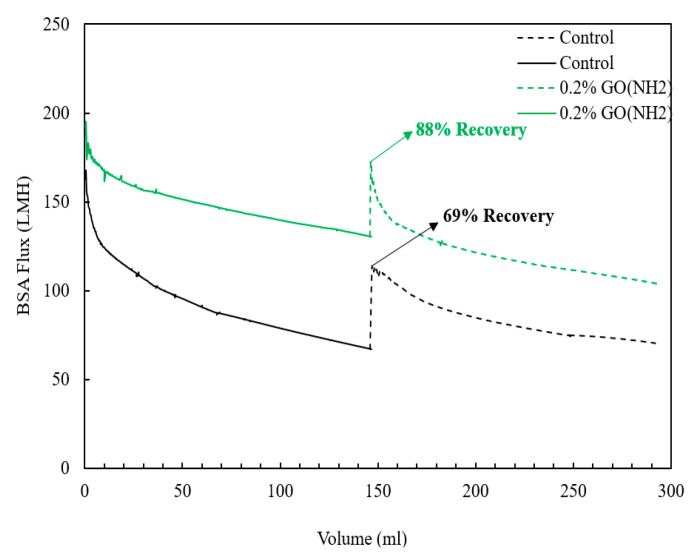
BSA fluxes vs. volume for control and 0.2 wt.% GO-NH_2_ membrane.

**Table 1 membranes-13-00678-t001:** Summarized results of control and optimum 0.2 wt.% GO-NH_2_ membrane.

Membrane	Contact Angle (°)	Porosity (%)	Young’s Modulus (MPa)	Breakage Strength (MPa)	Permeability (LMH/Bar)	Oil Rejection (%)	BSA Flux Recovery (%)
Control	77	67.5	60	2.2	1080	91.7	69
0.2% GO-NH_2_	65	68.9	117	3.3	2071	95.6	88

**Table 2 membranes-13-00678-t002:** Summarized results of air dehumidification performance of PSf and PSf- 0.2 GO-NH_2_ membranes.

Membrane	Absolute Humidity (kg/m^3^)		Humidity Reduction, %	Permeance (GPU)		Water/ Air Selectivity	Separation Factor
	Feed	Retentate		Water	Air		
PSf	0.0271	0.0191	29.4	13710	2020	7	1
PSf-0.2 GO-NH_2_	0.0280	0.0183	34.6	16410	2680	6	8

## Data Availability

Data will be available upon a reasonable request to the corresponding author.

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
