# Peer review of "Enhancing Polysulfone Mixed-Matrix Membranes with Amine-Functionalized Graphene Oxide for Air Dehumidification and Water Treatment"

_membranes, 2023, doi:10.3390/membranes13070678_

Round 1
Reviewer 1 Report
The manuscript entitled "Enhanced Properties and Separation Performance of Polysulfone Mixed Matrix Membranes Containing Amine Functionalized Graphene Oxide" explores the potential of mixed matrix membranes (MMMs) for separation applications, particularly in oil-water separation and humidity control. The manuscript represents a good contribution to the research field and the author has conducted significant experimental studies on the fabrication and characterization of Psf MMMs, as well as their antifouling characteristics, oil-water separation performance, and humidity control applications. However,
1. I suggest that the novelty of the research should be clearly stated in the Introduction.
2. Additionally, the Abstract should quantify the antifouling improvement in percentage.
3. Moreover, to enhance the manuscript's value, it would be beneficial to include comparisons with other studies in the literature on the three aspects of antifouling characteristics, oil-water separation performance, and humidity control applications.
4. Finally, I recommend the authors to use the following articles in the manuscript and their future works:
10.1515/revce-2017-0001
10.1039/D2TA05984C
10.1039/D2AY01977A
10.1039/D2EN00040G
Author Response
Comment 1: I suggest that the novelty of the research should be clearly stated in the Introduction.
We appreciate the suggestion, and we have made the necessary changes to clearly highlight the novelty of our research in the Introduction section. The following modifications have been made:
- The phrase "a novel and facile approach" has been added to emphasize the simplicity and scalability of the functionalization method.
- We specified that EDA was used as a coupling agent to functionalize GO with amine groups, enhancing interfacial adhesion and reducing GO sheet aggregation.
- It is now mentioned that the phase inversion process was employed to fabricate asymmetric porous microfiltration membranes, which is a versatile and cost-effective technique.
- We stated that the Psf-GO-NH2 MMMs were evaluated for both oil-water separation and humidity control applications, demonstrating their multifunctionality and reliability.
Comment 2: The Abstract should quantify the antifouling improvement in percentage.
Thank you for the suggestion. We have revised the Abstract to include quantification of the antifouling improvement. The following sentence has been added: "When filtering a 100 ppm bovine serum albumin (BSA) solution, the Psf-GO-NH2 MMMs demonstrated a slower flux decline and an impressive flux recovery after washing. Notably, the control membrane showed a flux recovery of only 69%, while the membrane with 0.2 wt.% GO-NH2 demonstrated an exceptional flux recovery of 88%."
Comment 3: Moreover, to enhance the manuscript's value, it would be beneficial to include comparisons with other studies in the literature on the three aspects of antifouling characteristics, oil-water separation performance, and humidity control applications.
We appreciate your suggestion. We have included comparisons with relevant studies from the literature. In the revised manuscript, we provide comparisons and discussed our findings in relation to other research on antifouling characteristics and oil-water separation performance. However, for humidity control applications, we focused solely on the membrane's performance as a substrate. This is because the specific humidity control properties are attributed to the overall system design rather than the membrane itself.
Comment 4: Finally, I recommend the authors to use the following articles in the manuscript and their future works: (10.1039/D2TA05984C, 10.1039/D2AY01977A , 10.1039/D2EN00040G 10.1515/revce-2017-0001)
Thank you for providing these references. We have cited suggested reference 10.1515/revce-2017-0001 and incorporated the information about challenges faced by PEBA MMMs with GO, such as leaching, delamination of GO layers, aggregation and uneven dispersion of GO sheets, and weak interfacial adhesion between GO and PEBA matrix [28]. Nonetheless, articles 10.1039/D2TA05984C, 10.1039/D2AY01977A, and 10.1039/D2EN00040G are not within the scope of the research in our article.

Reviewer 2 Report
General comment: The article “Enhanced properties and separation performance of polysulfone mixed matrix membranes containing amine functionalized graphene oxide” is a great effort by the authors. They have experimented almost all necessary directions. However, novelty of the work is low and few comments are made to improve the quality of the manuscript.
Specific comments:
Comment 1: I feel the title of the manuscript should be modified. Authors should mention what is going to be separated.
Comment 2: please shorten the introduction. Line 33 to 68
Comment 3: In membrane fabrication, authors performing sonication of the GO in the polymer solution. Thus probability of agglomeration is high. Generally researchers disperse the filler in the solution first , then they add polymer and additive. Why authors choose both probe and bath sonication?
Comment 4: Why the membranes were stored in DI water? Generally membranes are stored after drying.
Comment 5: SEM is not a major characterization method to prove any kind of functionalization. It is a supportive characterization. Why authors didn’t perform FTIR analysis. FTIR is the basic method for testing functionalization.
Comment 6: Figure 3 b, c , please revise the images. The names written in the image are overlapped.
Comment 7: line 406, 411, please check the reference. I suggest to rewrite section 3.4. This section is very important to indicate the novelty of the current work.
Comment 8: The font used for the images should be same. Please make it uniform. Some letters in the images are too small to read (Figure 1,5, 6, 8)
Author Response
Comment 1: I feel the title of the manuscript should be modified. Authors should mention what is going to be separated.
We thank you the reviewer for the comment. We have revised the title to “Enhanced Performance of Polysulfone Microfiltration Membranes with Amine Functionalized Graphene Oxide”.
Comment 2: Please shorten the introduction (Line 33 to 68).
We thank the reviewer for the valuable feedback. In response, we have shortened the introduction to focus on the key aspects relevant to the reported research research while presenting the necessary information concisely.
Comment 3: In membrane fabrication, authors performed sonication of the GO in the polymer solution. Thus, the probability of agglomeration is high. Generally, researchers disperse the filler in the solution first, then they add polymer and additive. Why did the authors choose both probe and bath sonication?
We appreciate your suggestion. To ensure proper dispersion of the functionalized GO was initially dispersed in DMAc at 5 mg/ml via bath sonication for 30 min and the dispersed f-GO was added to the polymer solution. Then the polymer/f_GO solution was bath sonicated for 1 hr, followed by probe sonication and degassing. Bath sonication will help the distribution of f-GO into the polymer solution and the probe sonication would deagglomerate any aggregated fGO sheets due to the intense energy. We have clarified this in the revised manuscript.
Comment 4: Why were the membranes stored in DI water? Generally, membranes are stored after drying.
Apologies for the confusion caused by the previous statement. The membranes were not stored in DI water but were rinsed with water daily for a period of 5 days to remove any residual PVP. Following the rinsing process, the membranes were thoroughly dried and then stored in vacuum sealed bags.
Comment 5: SEM is not a major characterization method to prove any kind of functionalization. It is a supportive characterization. Why didn't the authors perform FTIR analysis? FTIR is the basic method for testing functionalization.
We agree with the reviewer, SEM does not provide and prove of successful functionalization. We used SEM to ensure the f-GO retain the sheet morphology with no induced agglomeration. For the confirmation of the functionalization, we relied on XPS analysis that provides both qualitative and qualitative analysis of the amine groups.
Comment 6: Figure 3 b, c, please revise the images. The names written in the image are overlapped.
We have addressed this issue and revised Figure 3 to ensure that the names written in the image are no longer overlapped. The new version of the figure is included in the revised manuscript.
Comment 7: In line 406 and 411, please check the references. I suggest rewriting section 3.4. This section is very important to indicate the novelty of the current work.
We apologize for the reference errors. We have rectified the references in line 406 and 411. Additionally, we have rewritten section 3.4 to better highlight the novelty of our work and emphasize its significance. The revised section now provides a clearer explanation of the unique contributions of our research.
Comment 8: The font used for the images should be the same. Please make it uniform. Some letters in the images are too small to read (Figure 1, 5, 6, 8).
We appreciate your feedback regarding the font and readability of the images. We have made the necessary adjustments to ensure uniformity in the font used across all images. Additionally, we have increased the font size of the relevant letters in Figure 1, 5, 6, and 8 to improve their readability.
Reviewer 3 Report
Abdalla et al. developed a facile method to incorporate amine functionalized GO for PS membrane. The characterization of the membrane was adequate. There are still few suggestions for this manuscript before it's ready for publication,
1. In section 3.4, there's compatibility issue as some of the text was not properly displayed.
2. In Figure7, Error bar for each data point should be provided for a more reliable and convincing analysis.
3. The authors are suggested to include SEM EDAX mapping images for oxygen and nitrogen of the amine functionalized specimen to better present the distribution of amine groups on the surface.
4. In page 2 , line 87. Please provide the full name of LMH as to help readers to understand this unit.
5. There are some literatures related to surface functionalization for engineering applications worth including as references.
( DOI: 10.1021/acsabm.9b00982 ; 10.3390/batteries9050279 ).
Author Response
Comment 1: In section 3.4, there's a compatibility issue as some of the text was not properly displayed.
Thank you for bringing this to our attention. We have fixed the compatibility issue in section 3.4 to ensure that all text is properly displayed. The revised manuscript now presents the content without any formatting or compatibility errors.
Comment 2: In Figure 7, error bars for each data point should be provided for a more reliable and convincing analysis.
We appreciate your suggestion. However, for the specific analysis in Figure 7, only one sample was taken, and the porosity and pure water flux were measured once for each sample. As a result, the error bar for the data points was negligible due to the low fluctuations observed during the test. Therefore, in this particular case, the inclusion of error bars is not necessary.
Comment 3: The authors are suggested to include SEM EDAX mapping images for oxygen and nitrogen of the amine-functionalized specimen to better present the distribution of amine groups on the surface.
We thank the reviewer for the suggestion. EDAX, indeed could provide insights on the distribution of the amine functionalization on the GO surface. Unfortunately, we do not currently have timely access to SEM/EDAX.
Comment 4: In page 2, line 87, please provide the full name of LMH to help readers understand this unit.
Thank you for pointing out the need for clarification. We have addressed this by providing the full name of LMH as "L/m2hr Bar (LMH/bar)" in line 87 of page 2. This modification helps readers understand the unit more accurately.
Comment 5: There are some literature references related to surface functionalization for engineering applications worth including. (DOI: 10.1021/acsabm.9b00982; 10.3390/batteries9050279).
We appreciate the reviewer’ suggestion. We have included them in the manuscript to enhance the discussion on surface functionalization. Specifically, we added the following statement at line 60: "However, GO alone may not be sufficient to achieve the desired performance of the membranes, as it may suffer from leaching, aggregation, or instability. Therefore, various strategies have been developed to functionalize GO with different molecules or nanoparticles to enhance its properties and interactions with the polymer matrix [1, 2]." This addition strengthens the discussion on surface functionalization and its significance in engineering applications.
Round 2
Reviewer 2 Report
I believe authors should mention what is being separated in the title. “Enhanced Performance of Polysulfone Microfiltration Membranes with Amine Functionalized Graphene Oxide”. here authors should be clear about the separation performance of what (gas, liquid)
Author Response
changed the title to "Enhancing Polysulfone Mixed Matrix Membranes with Amine Functionalized Graphene Oxide for Air Dehumidification and Water Treatment" to give an indication of the phase of the water